# Factors and Pathways Modulating Endothelial Cell Senescence in Vascular Aging

**DOI:** 10.3390/ijms231710135

**Published:** 2022-09-04

**Authors:** Hyun Jung Hwang, Nayeon Kim, Allison B. Herman, Myriam Gorospe, Jae-Seon Lee

**Affiliations:** 1Research Center for Controlling Intercellular Communication, College of Medicine, Inha University, Incheon 22212, Korea; 2Department of Molecular Medicine, College of Medicine, Inha University, Incheon 22212, Korea; 3Program in Biomedical Science and Engineering, College of Medicine, Inha University, Incheon 22212, Korea; 4Laboratory of Genetics and Genomics, National Institute on Aging-Intramural Research Program, NIH, Baltimore, MD 21224, USA

**Keywords:** cellular senescence, endothelial cell, molecular player, signaling pathway, putative target, age-related vascular disease

## Abstract

Aging causes a progressive decline in the structure and function of organs. With advancing age, an accumulation of senescent endothelial cells (ECs) contributes to the risk of developing vascular dysfunction and cardiovascular diseases, including hypertension, diabetes, atherosclerosis, and neurodegeneration. Senescent ECs undergo phenotypic changes that alter the pattern of expressed proteins, as well as their morphologies and functions, and have been linked to vascular impairments, such as aortic stiffness, enhanced inflammation, and dysregulated vascular tone. Numerous molecules and pathways, including sirtuins, Klotho, RAAS, IGFBP, NRF2, and mTOR, have been implicated in promoting EC senescence. This review summarizes the molecular players and signaling pathways driving EC senescence and identifies targets with possible therapeutic value in age-related vascular diseases.

## 1. Introduction

Cellular senescence is a state of indefinite cell cycle arrest implemented in response to sublethal stresses, such as DNA damage, telomere shortening, oxidative stress, and oncogene activation. The implementation of senescence involves dynamic changes in protein expression programs, as well as morphology and metabolism. These changes are coordinated by different signaling pathways, some of them culminating with the activation of p53 and a rise in the production of cyclin-dependent kinase (CDK) inhibitors, such as p16 (CDKN2A), p15 (CDKN2B), p21 (CDKN1A), and p27(CDKN1B) [1]. Inhibition of CDK-cyclin complexes is important to suppress growth through the hypophosphorylation of pRb (RB) [2]. Cellular senescence is beneficial for the organism during embryonic development, tissue morphogenesis, wound healing, and tumor suppression in young persons [1,2,3,4,5,6]. However, in older persons, the increased burden of senescent cells has many detrimental actions, exacerbating age-related diseases, including cancer, neurodegeneration, and cardiovascular disease.

A blood vessel consists of three layers, the tunica intima (inner layer), the tunica media (middle layer), and the tunica adventitia (outer layer). The tunica intima is a barrier between the vessel and blood flow and is composed of a layer of endothelial cells and an internal elastic lamina. Endothelial cells (ECs) play a significant physiological role in vascular homeostasis, maintaining blood flow, and regulating vascular tone, pro-inflammatory responses, and neovascularization [7]. Since ECs come into direct contact with the bloodstream, EC senescence can also be triggered by mechanical stimuli, such as disturbed flow [8,9]; although the molecular mechanisms are not completely characterized, disturbed flow induces DNA damage, telomere disfunction, and production of excessive reactive oxygen species (ROS) in ECs [9]. Senescent ECs generally exhibit typical traits of senescent cells, such as a flattened and enlarged morphology, increased polyploidy, decreased nitric oxide (NO) bioavailability, and secretion of a myriad of pro-inflammatory cytokines [10,11,12]. Consequently, endothelial senescence promotes the pathogenesis of vascular aging by decreasing vascular density, increasing intima-media layer thickness and collagen deposition, reducing elastin deposition, and expanding the vascular lumen [13,14,15,16]. As a result of these physiological changes, the vessel experiences increased vascular stiffness caused by decreased arterial elasticity, enhanced vascular inflammatory responses, in turn leading to impaired angiogenesis and vascular tone [8,17]. These alterations in aged vessels are significantly associated with age-related vascular diseases, such as atherosclerosis and hypertension.

Given that senescent ECs play crucial roles in age-related pathophysiological changes in the vessel wall, understanding the mechanisms that promote EC senescence is essential for developing therapeutic strategies to combat age-related vascular disease. This review focuses on the molecular mechanisms and emerging therapeutic targets of EC senescence and age-related vascular pathologies.

## 2. Characteristics of Endothelial Cell Senescence

Senescent ECs display an enlarged and flattened morphology, a classic phenotype of senescent cells. In addition, senescent ECs exhibit increased polyploidy, senescence-associated-β-galactosidase (SA-β-Gal) activity, and telomere shortening [18,19,20,21,22]. In addition to classical senescence markers (e.g., p16, p21, and phosphorylated (phospho)-p38), other changes have been identified in senescent ECs, including higher production of fibronectin, intercellular adhesion molecule 1 (ICAM-1), and inducible nitric oxide synthase (iNOS) (Figure 1) [23,24,25,26,27].

Senescent ECs produce a complex senescence-associated secretory phenotype (SASP) that varies depending on the type, duration, and intensity of damage, as well as on EC growth conditions [28]. For instance, replicatively senescent HUVECs show increased production of interleukin 8 (IL8), IL6, plasminogen activator inhibitor 1 (PAI-1 or SERPINE1), and monocyte chemotactic protein-1 (MCP-1/CCL2), and IL-1α (IL1A) [29,30,31]. By contrast, HUVECs rendered senescence by disturbed flow show moderate ROS production, which is a distinct SASP feature observed in replicatively senescent HUVECs [32]. In another example, exposure to chronic TNF-α (TNF) induces HUVEC senescence with production of many cytokines, such as IL6, IL8, PAI-1, and IGFBP5 (insulin-like growth factor binding protein-5) [33]. Other triggers of HUVEC senescence, including exposure to radiation or doxorubicin (Doxo), led to increases in the level of the C–X–C motif chemokine 11 (CXCL11), which promotes the aggressive activity of cancer cells by binding to the C–X–C motif chemokine receptor 3 (CXCR3) [34].

Compared with proliferating HUVECs, senescent HUVECs have extended and interconnected mitochondria associated with reduced expression of FIS1 and DRP1, two important proteins in mitochondrial fission [35]. Silencing DRP1 induces senescence in young HUVECs with increased SA-β-Gal staining, elongated mitochondria, impaired autophagy, and increased expression of p21 and p16. DRP1 abundance is reduced in the endothelium of aorta in old rats and is associated with impaired autophagy. Silencing DRP1 also impairs autophagic flux in the vascular endothelium of carotid arteries in rats [30] and excessive mitochondrial fission induces EC senescence [36]. Loss of protein disulfide isomerase A1 causes EC senescence by inducing the production of mitochondrial ROS and mitochondrial fragmentation through the DRP1 sulfenylation at cysteine 644 [36].

Oxidative stress is a well-known contributor to EC senescence [37]. ROS is not only generated through mitochondrial respiration, but also from peroxisomal β-oxidation of free fatty acids, xanthine oxidase, lipoxygenase, nicotinamide adenine dinucleotide phosphate (NADPH) oxidase (NOX), microsomal P-450 enzymes, cyclooxygenases, and prooxidant heme molecules [38]. Unlike mitochondria in other cells, EC mitochondria are not considered as the primary source of ROS generation, which could be associated with the small ratio of mitochondria to EC by mass [39]. Instead, nicotinamide adenine dinucleotide phosphate (NADPH) oxidases are considered to be the main source of ROS formation in ECs; additional sources include the lysosome, peroxisome, and endoplasmic reticulum (ER) [40,41,42,43]. NOX is one of the key enzymes responsible for much of the generation of ROS in cardiovascular disease (CVD). NOX1, NOX2, NOX4 and NOX5 isoforms have been found to induce EC dysfunction, inflammation and apoptosis in atherosclerosis, hypertension, and diabetes [44]. Although physiological ROS levels are required to maintain normal cellular function, overproduction of ROS induces deleterious effects, such as changes in DNA transcription, interruption of many redox-sensitive signaling pathways, impairment of cellular structure and function, inflammation, and organ dysfunction [45]. In late-passage ECs, a rise in the level of intracellular ROS further contributes to EC senescence by accelerating telomere shortening [46,47].

Senescent cells are metabolically active and generally require more energy from glycolysis. However, senescent ECs undergo senescence-associated metabolic shifts, with decreased glycolysis and oxidative phosphorylation [37,48,49,50]. A recent report [51] found that senescent cells increase the production of glutaminase and consume high levels of glutamine, in turn producing glutamate and ammonia. This process is believed to enhance senescent cell survival by helping to maintain internal cellular pH; accordingly, inhibiting glutaminolysis was found to selectively remove senescent ECs and alleviated age-associated organ dysfunction [51]. Investigation of glutaminolysis on the EC metabolism under different disease conditions may provide new targets to manage EC senescence. Dysfunctional ECs show a higher dependence on oxidative phosphorylation (OXPHOS) than on glycolysis [52]. Although OXPHOS yields more ATP than glycolysis, high levels of OXPHOS lead to an increase in ROS and EC damage [53]. Hyperglycemic stress markedly reduces glycolytic enzymes and shifts glycolysis to OXPHOS in the diabetic endothelium [52]. Haspula et al. found that angiogenic signaling was compromised in the presence of a high superoxide burden, possibly due to the rise in OXPHOS activity [52].

Senescent cells have been observed in a variety of mammalian vascular tissues associated with cellular stress, aging, and age-related pathologies. For example, single and double balloon denudations of rabbit carotid arteries caused EC senescence, as identified by SA-β-gal staining; the authors suggested that vascular cell senescence could affect atherogenesis and post-angioplasty restenosis [54]. In addition, senescent ECs were found in Zucker diabetic rats at six-fold higher levels by 22 weeks of age when compared with control rats, with dramatic increases in p53, p21, and p16 levels [55,56]; the number of SA-β-Gal-positive ECs declined after treatment with ebselen, a peroxyni-trite scavenger [56]. The aging process alone can increase senescence, as a single-cell RNA sequencing analysis found that ~10% of ECs in the brain microvasculature of 28-month-old mice were senescent [57]. Similarly, SA-β-gal-positive ECs exist in human atherosclerotic plaques of the aorta, coronary arteries, and adipose tissues of obese human subjects [26,58,59]. Taken together, these studies illustrate the prevalence and importance of senescent ECs in age-related vascular diseases.

## 3. Molecular Players and Pathways Associated with Endothelial Cell Senescence

Many molecular mechanisms have been implicated in endothelial and vascular cell senescence. Those best characterized are described below and summarized in Figure 2.

### 3.1. Sirtuins

Sirtuins comprise a family of signaling proteins with metabolic functions, particularly nicotinamide adenine dinucleotide (NAD+)-dependent deacetylase and ADP-ribosyltransferase activities [60]. Among the seven sirtuins, SIRT1 is particularly highly expressed in ECs of arteries, veins, and capillaries, and plays crucial roles in EC senescence, DNA repair, cell cycle regulation, gene silencing, stress resistance, apoptosis, organism longevity, and inflammation in the vascular system [60]. Dysfunction of the SIRT1–eNOS axis results in impaired NO production during the aging-associated downregulation of SIRT1 [61,62,63]. Furthermore, the protection against ROS-induced premature senescence by SIRT1 was linked to an increase in eNOS activity, as supported by observations that an endogenous suppressor of SIRT1 production, miR-217, suppressed eNOS activity and advanced endothelial senescence [62,63].

During inflammatory processes underlying the initiation and progression of vascular aging, ECs and activated immune cells exhibit chronically active NF-κB signaling. In this paradigm, reductions in the levels of SIRT1 lead to increased acetylation of Lys-310 on the p65 subunit of NF-κB, resulting in an increased inflammatory response in ECs and monocytes/macrophages [64,65,66]. In ECs, increased CDK5-mediated hyperphosphorylation of SIRT1 at Ser-47 prevents the nuclear export of SIRT1 and its interaction with telomeric repeat-binding factor 2-interacting protein 1 (TERF2IP), a regulator of telomere function and NF-κB signaling [67]. Other factors blocked by SIRT1, including p53, PAI-1, and p66^Shc^, participate in the protection against endothelial senescence [68,69,70]. Moreover, in aged mice, a decline in SIRT1 activity, due to impaired phosphorylation at Ser-154, accounts for the reduced expression of endothelium estrogen receptor β (ERβ) via binding of SIRT1-PPAR-c/RXR-p300 to a PPAR response element site on the ERβ promoter [71]. Consistent with the role for SIRT1 in endothelial senescence, a mouse model of vascular senescence produced by genetically excluding exon 4 of SIRT1 in ECs (*SIRT1*^endo−/−^) exhibited damaged endothelium-dependent angiogenesis and vasorelaxation [72].

More recently, another molecular mechanism underlying the protective effect of SIRT1 against endothelial senescence and vascular aging was reported. SIRT1 was found to prevent endothelial senescence by reducing the acetylation of stress-responsive serine/threonine liver kinase B1 (LKB1) via HERC2, a giant scaffold protein and E3 ubiquitin ligase [73]. In senescent ECs and aged arteries, loss of SIRT1 caused acetylated LKB1 to accumulate in the nucleus, with irreversible changes in the blood vessel wall, vascular stiffness, and adverse arterial remodeling. These changes suggest that the SIRT1/HERC2/LKB1 regulatory paradigm fine-tunes the communication between endothelial and vascular smooth muscle cells (VSMCs) to maintain vascular homeostasis [73]. Conversely, SIRT1 overexpression and increased LKB1 deacetylation prevent EC senescence in culture and stress-induced senescence in mice [70].

SIRT6, a chromatin-associated protein that stabilizes genomes and telomeres, protects cells from premature senescence by maintaining their replicative capacity and angiogenic ability in culture [74]. In ECs, SIRT6 was found to decrease during H_2_O_2_-induced senescence, while overexpression of SIRT6 partially reversed this process [75]. Knockdown of SIRT6 accelerated cell senescence and overactive NF-κB signaling, suggesting that SIRT6 plays a fundamental role in aging and inflammation [76]. SIRT6 prevents the formation of volatile atherosclerotic plaques in diabetic patients, and silencing SIRT6 suppresses EC replication and promotes EC senescence [77,78]. In addition, SIRT6 was proposed to inhibit heart failure and cardiac hypertrophy by regulating signaling through the insulin/insulin-like growth factor (IGF)-Akt [79]. SIRT6 was also postulated to protect hepatic low-density lipoprotein (LDL) receptors from degradation and reduced plasma LDL cholesterol levels in mice by inhibiting the transcriptional increase in proprotein convertase subtilisin/kexin type 9 (PCSK9), both of which may prevent atherogenesis [80]. Overall, SIRT1 and 6 engage in critical functions to maintain vascular function and integrity with age.

### 3.2. Klotho

Klotho is expressed primarily in the kidney and, to a lesser extent, in the vascular tissue [81]. Animal studies offer evidence of the protective capacity of Klotho in endothelial dysfunction via the upregulation of NO [82]. In support of this role, mice with ablated *Klotho* gene exhibit damaged endothelium and NO-dependent vasodilation in aorta and arterioles from when exposed to acetylcholine [83]. Klotho was also found to promote endothelial health by inducing anti-oxidative pathways; for example, it decreased H_2_O_2_-induced apoptosis in HUVECs by promoting manganese superoxide dismutase (MnSOD) activity through insulin/IGF signaling. Besides promoting resistance to oxidative stress, Klotho has anti-aging effects by suppressing caspase activity, reducing signaling through the p53/p21 pathway, and inducing resistance to oxidative stress [83,84,85,86].

Clinical trials showed that adults with high level of plasma Klotho have lower risk of coronary artery disease (CAD), heart failure, stroke, and peripheral arterial disease (PAD) [87]. Conversely, lower Klotho levels in serum correlated with the presence and severity of CAD separately from other well-known cardiovascular disease (CVD) risk factors, such as dyslipidemia, diabetes, and hypertension [88]. Furthermore, the level of Klotho in serum was an independent predictor of arterial stiffness [89]. Until now, overexpression of Klotho prevented renal and cardiovascular dysfunction related to aging and extended lifespan in experimental mice [86]. *Klotho* knockout mice showed premature aging characteristics, including changed calcium/phosphate metabolism, vascular calcification, and decreased lifespan [90]. These studies generally support the hypothesis that high levels of soluble Klotho may protect ECs and cardiovascular functions from age-related dysfunction, and extend human lifespan.

### 3.3. Renin-Angiotensin-Aldosterone System

The renin-angiotensin-aldosterone system (RAAS) is a well-known regulator of salt and water homeostasis. Plasma renin converts angiotensinogen to angiotensin I, which is then converted by angiotensin-converting enzyme (ACE) on the surface of vascular endothelial cells to angiotensin II (Ang II) [91]. Angiotensin (Ang) II is a significant effector of the renin-angiotensin-aldosterone system (RAAS) and functions in controlling cardiovascular hemodynamics and structures, serving as a mitogenic factor and a vasoconstrictor [92]. Ang II acts through two main cell surface receptor subtypes, Ang II type 1 and Ang II type 2 receptors (AT1R and AT2R, respectively). The mitogenic function of Ang II is linked to initiating and maintaining the G1-S phase transition in the cell cycle, which is mediated through the AT1R [92,93]. Ang II increases ROS production, inflammation, extracellular matrix remodeling, and vessel tone through AT1R [10]. Ang II also promotes senescence of ECs, associated with age-related vascular impairments in humans and rodents [94,95,96], possibly linked to the presence of ECs with senescence-associated features in human atherosclerotic lesions [26]. Ang II induces EC senescence and inflammation, at least partly via the activation of MAPK [97,98]. Interestingly, AT1R blockers suppress Ang II-induced EC senescence and have a protective effect against the age-related vascular diseases, such as hypertension and atherosclerosis in rodents [99,100]. These results suggest that Ang II promotes EC senescence through AT1R-coupled signaling mechanisms. A study found that expressions of Ang II, AT1R, NOX2, collagen IV, and fibronectin increased, while the expression levels of the G protein-coupled receptor Mas and AT2R decreased in 24-month-old mice [101]. The heptapeptide Ang-(1-7), a key player of the so-called protective branch of the RAAS [102], counteracts the Ang II-induced senescence of cultured human ECs by acting upon Mas and attenuating the increase in SA-β-Gal [103]. Ang-(1-7) further reduces total and telomeric DNA damage, an early event that releases signals downstream to trigger growth arrest and senescence [103]. The anti-senescence action of Ang-(1-7) is further supported by evidence that p53 was markedly attenuated by the co-infusion of Ang-(1-7) in the aortas of C57BL6/J mice infused with IL-1β [104]. Overall, these findings underscore the pleiotropic benefits of the Ang-(1-7)/Mas receptor axis beyond Ang II and the RAAS, and reinforce the potential interest of Ang-(1-7) in suppressing EC senescence [105].

### 3.4. Insulin-like Growth Factor-Binding Proteins

Insulin-like growth factor (IGF)-binding protein 1 (IGFBP1) is a member of the IGFBP family, which consists of six structurally similar proteins (IGFBP1 through ~6) with strong affinity for IGF. IGFBP1 participates in metabolic homeostasis and cell growth [105,106]. Activation of the IGF1 receptor by IGF1 and IGF2 mediates radiation-induced senescence of human pulmonary artery endothelial cells (HPAECs) [107]; paradoxically, however, IGF1 prevents the onset of senescence by oxidative stress in human aortic endothelial cells (HAECs) [108]. IGFBP3 is upregulated in replicatively senescent ECs in culture [109]. The senescent phenotype of HUVECs was enhanced by IGFBP3 overexpression and was suppressed by IGFBP3 silencing [110]. IGFBP5 induced senescence in HUVECs through a p53-dependent mechanism and is a potential candidate for radiation-induced senescence in HUVECs [111,112,113].

### 3.5. Nuclear Factor-E2-Related Factor

Nuclear factor-E2-related factor 2 (NRF2) is a transcription factor sensitive to oxidative stress. NRF2 binds to antioxidant response elements (AREs) in the nucleus and enhances the transcription of several antioxidant genes [114]. Activation of NRF2 was found to prevent the induction of cellular senescence [106], and genetic ablation of NRF2 was found to aggravate cell senescence in older cerebral arteries, associated with increased expression of inflammatory cytokines in vessels and hippocampus [115]. Moreover, the effects of cardiovascular risk factors (e.g., obesity) were aggravated, at least partly due to accelerated cellular senescence, in genetically *Nrf2*-deficient mice [116,117,118]. Previous studies, including studies in non-human primates, reported that decreased NRF2-mediated antioxidant response and suppression of mitochondrial SOD2 induced chronic oxidative stress in EC senescence [119]. Ungvari et al. demonstrated that the carotid arteries of aged *Rhesus* macaques showed substantial oxidative stress compared to the vessels of younger animals [120]. Old monkeys showed impaired nuclear translocation of NRF2 and reduced expression of target genes *NQO1*, *GCLC,* and *HMOX1*. Moreover, in response to H_2_O_2_ and high-glucose conditions, VSMCs from young monkeys showed considerably higher expression of NRF2-regulated genes, while activation of the NRF2 signaling pathway was suppressed in VSMCs of aged monkeys [120]. Similar results were obtained in aged rats, which showed dampened NRF2 activity and expression of NRF2 target genes, and elevated expression of the NF-κB target genes in the vasculature [121]. In sum, senescence impairs NRF2 function, contributing to a vicious cycle that aggravates cellular damage induced by senescence-associated oxidative stress.

### 3.6. Mammalian Target of Rapamycin

Mammalian target of rapamycin (mTOR), a serine/threonine kinase of the phosphatidylinositol-3-OH kinase (PI3K)-associated family, is a core component of two distinct multi-protein complexes, mTORC1 and mTORC2 [122,123]. In HUVECs, alleviation of mTORC1 signaling with low doses of rapamycin delayed replicative senescence and increased proliferation and tube formation [124]. Similarly, treatment of primary human fibroblasts and fibroblasts that overexpressed the RAS oncogene with rapamycin reversed replicative and oncogene-induced senescence, due to the inhibition of mTORC1 activation. Mechanistically, rapamycin inhibits S6 kinase (S6K) and reduces the phosphorylation of the ribosomal subunit S6 at Ser 235 in senescent cells [125]. In aged mice under caloric restriction (CR), blockade of mTOR signaling enhanced endothelium function; these effects were linked to the CR-mediated reduction in aortic mTOR activation and the protection of age-associated arterial endothelium-dependent dilation (EDD) by increasing NO and preventing the superoxide-dependent suppression of EDD [126,127]. Similarly, treatment of old Wistar Kyoto (WKY) rats with resveratrol or rapamycin also suppressed ribosomal protein S6 kinase 1 (S6K1) activation and enhanced NO generation by endothelial cells [128], while inhibiting the PI3K/AKT/mTORC1 pathway in differentiating mouse embryonic stem cells increased vascular EC elongation triggered by VEGF [129]. In agreement with the fact that mTOR modulates SASP factor production [130,131,132], inhibiting the PI3K/AKT/mTOR pathway in Doxo-treated HUVECs was found to induce a senescence secretory response different from the canonical SASP response [133]. A self-limiting SASP response, mediated through the inhibition of PI3K/AKT/mTOR pathway, suppressed strong senescence-associated inflammation and served to maintain tissue homeostasis [133]. While many studies have explored the function of mTORC1 in aging and longevity, much less is known about the function of mTORC2. Yang et al. recently reported that mTORC2 activity increases during replicative senescence and H_2_O_2_-induced premature senescence in HUVECs and found that silencing mTORC2 and inhibiting AKT regulated NRF2 and alleviated HUVEC senescence [134].

### 3.7. Others Molecules

Many other endothelial senescence-related molecules have been identified. Fibro-blast growth factor 21 (FGF21) prevents high glucose-induced cellular damage and in-hibition of eNOS activity in cultured HUVECs [135]. FGF21 protects Ang II-induced cerebrovascular aging through inhibiting p53 activation and inducing mitochondrial biogenesis in an AMP-activated protein kinase (AMPK)-dependent manner [136]. The adaptor protein P66^shc^ is a signaling molecule involved in the pathways that regulate redox metabolism [137]. In aortic rings from young and old mice, P66^shc^ was implicated in protecting against an age-dependent impairment of EC relaxation in response to acetylcholine [138]. Guidance receptor UNC5B promotes EC senescence, potentially by activating the ROS-p53 pathway [139]. IL-17A is an inflammatory cytokine produced by Th17 cells (a subgroup of helper T cells), which contributes to EC senescence via activation of the NF-κB/p53/RB signaling pathway [140]. Peroxisome proliferator-activated receptor alpha (PPARα) inhibits EC senescence by promoting the expression of the aging-related protein growth differentiation factor 11 (GDF11), the levels of which decline with age in several organs in mice [141]. Deficiency in developmentally regulated GTP-binding protein 2 (DRG2) increases NOX2 expression, ROS generation, and senescence in ECs [142]. C1q/tumor necrosis factor-related protein 9 (CTRP9), an emerging potential cardiokine, contributes to the inhibition of hyperglycemia-induced endothelial senescence through an AMPKα/ Krüppel-like factor 4 (KLF4)-dependent signaling pathway [143]. Soluble dipeptidyl peptidase 4 (DPP4), secreted from visceral adipose tissue, induces EC senescence through the protease-activated receptor 2 (PAR2)–cyclooxygenase2 (COX-2)–thromboxane receptor (TP) axis and activates the NLRP3 inflammasome [144]. In cultured HAECs, senescence induced by apoC3-rich low-density lipoprotein (AC3RL) is mediated by intracellular ROS formation, H2A.X variant histone (H2AX) deposition, and F-box only protein 31 (FBXO31) activation, resulting in the inhibition of mouse double minute 2 homolog (MDM2), p53, and p21 activation [145]. Resolvin E1 (RvE1) reduces EC senescence induced by both Doxo and IL-1β through preventing the increase in levels of pP65, NLRP3, and pro-IL-1β and the formation of active NLRP3 inflammasome complexes [146]. While we have highlighted the numerous molecular players and pathways involved in EC senescence, we expect that many more will be discovered as we continue to dissect the mechanisms that foment senescence in vascular diseases.

## 4. Conclusions and Perspectives

EC senescence is induced in many different ways, including telomere shortening and damage to DNA and other cell components [147]. Increasing evidence shows an important role for EC senescence in vascular aging and age-related vascular disease. To develop therapeutic targets to treat or prevent vascular aging and age-related vascular disease, we must continue to elucidate the molecular mechanisms that promote EC senescence in cell and animal models. SIRT1 and SIRT6 prevent EC senescence by blocking oxidative stress-induced signaling. Interestingly, while Klotho and NRF2 prevent EC senescence, other factors, such as angiotensin, IGFBP3, IGFBP5, and mTOR, promote EC senescence. A comprehensive identification of druggable targets in this paradigm will broaden the therapeutic opportunities to attenuate EC senescence and improve vascular disease outcomes. In Ang II-induced senescence, olmesartan, a pharmacological inhibitor of Ang II, suppressed Ang II-induced vascular inflammation and premature senescence [95]. The identification of novel markers and improved methods to detect senescent ECs in vivo systems are necessary to accelerate progress towards finding more efficient ways to discover senescent ECs in various tissues and organs. Further studies focused on the molecular mechanisms and signaling pathways of EC senescence will accelerate the development of strategies to prevent age-related vascular disease.

## Figures and Tables

**Figure 1 ijms-23-10135-f001:**
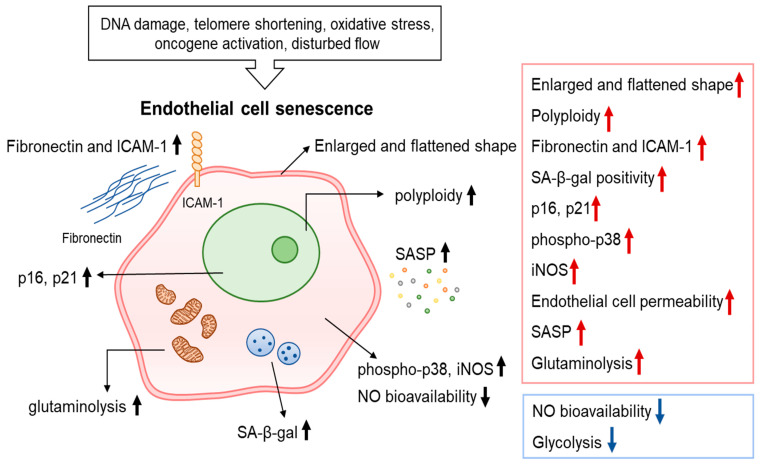
Characteristics of senescent endothelial cells. Various stimuli including DNA damage, telomere shortening, disturbed flow induce endothelial cell (EC) senescence. EC senescence is a state of indefinite cell cycle arrest accompanied by multiple biochemical and metabolic changes. Senescent cells generally become enlarged and flattened and display increased polyploidy, SA-β-gal activity, permeability, phosphorylation of p38, and expression of p16/p21. In addition, senescent ECs display elevated fibronectin, ICAM-1, and iNOS and reduced of NO bioavailability. Senescent ECs also produce a complex and unique SASP that includes secretion of IL-8, IL-6, PAI-1, MCP-1, IGFBP-5, and CXCL11. The metabolic alterations of senescent ECs include reduced glycolysis and increased glutaminolysis.

**Figure 2 ijms-23-10135-f002:**
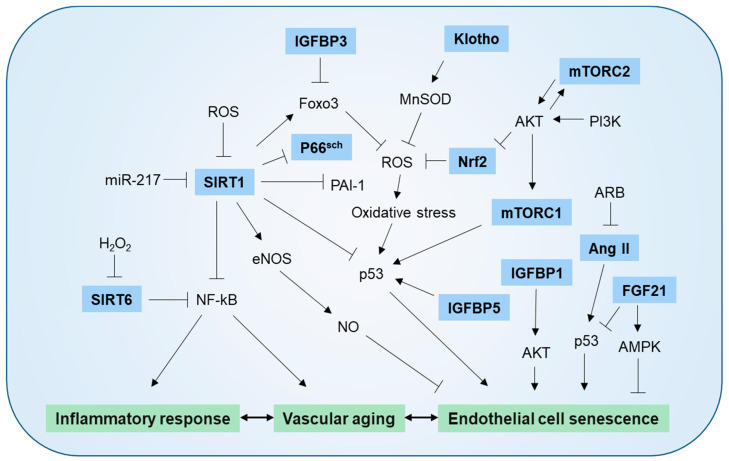
Molecular mechanism mediating endothelial cell senescence and vascular aging. SIRT1 protects against endothelial dysfunction by deacetylating eNOS and increasing endothelial NO bioavailability. SIRT1 also protects vascular inflammation by suppressing NF-κB signaling and exerts anti-oxidant action by increasing Foxo and down-regulating p66shc. Under conditions of age-related overexpression, miR-217 suppresses SIRT1 production. Klotho increases MnSOD activity and NO availability by negatively downregulating IGF-1 signaling. H2O2 induces endothelial cell (EC) senescence via downregulation of SIRT6 levels. SIRT6 deficiency raises the expression of endothelial proinflammatory cytokines and increases NF-κB transcriptional activity. Increased oxidative stress due to NRF2 deficiency promotes EC senescence. mTORC1 increases EC senescence through the PI3K/AKT pathway and mTORC2 induces EC senescence by suppressing NRF2 production via the AKT/GSK-3β/CEBPα signaling pathway. IGFBP1 and IGFBP5 induce senescence in HUVECs through AKT or p53-dependent mechanism. IGFBP3 promotes EC senescence by inhibiting the SIRT1-Foxo axis, ARB inhibits Ang II-induced EC senescence, and FGF21 represses EC senescence via inhibiting p53 signaling pathway in an AMPK-dependent manner.

## Data Availability

Not applicable.

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
