# Peer review of "Factors and Pathways Modulating Endothelial Cell Senescence in Vascular Aging"

_ijms, 2022, doi:10.3390/ijms231710135_

Round 1
Reviewer 1 Report
Major Comments:
- The abstract should improve. It is ambiguous and hard to focus.
- This review manuscript tries to cover so many contents, however, there is a lack of depth and recent findings. It is better to focus on fewer topics.
- In therapeutic targets related to vascular disease sections, most of the sections just describe the role of ECs in each disease. Those are nothing new or a lack of connection between the pathways described in section 3 and cardiovascular diseases.
- Aging and senescence are not the same. But the authors used them without distinction. Please be careful to use the term.
Minor Comments:
- In the abstract, it is hard to understand “With advancing age, an accumulation of vascular endothelial cells contributes to …” sentence. Do you mean the number of ECs is increased with aging? and…does it cause cardiovascular diseases like hypertension, atherosclerosis, and etc?
- In line 45, “ECs have a limited lifespan…”. This is not an EC-specific phenotype. Most of cell types except cancer cells have a finite lifespan. One of the EC phenotypes is that EC turnover time is relatively longer than other cell types. It would be better to discuss this phenotype.
- In line 46-49, this sentence is not really correct. There is a consensus that nitric oxide bioavailability is reduced in senescent or aged ECs, whereas eNOS levels are still controversial in those ECs. If you want to describe eNOS levels please provide several references. Also, references 10 and 11 are not appropriate references.
- Line 52, “physiological” should be replaced by morphological.
Reviewer 2 Report
The review by Hwang et al explored the mechanisms of endothelial senescence. It covers most of the mechanisms, and lists some well-known targets and some that have not been detailed as well in other reviews. It is a well-written and well-organized.
However the review is not fleshed out as much. This leads to some very important sections missing key details for the readers to fully appreciate the scope of endothelial senescence.
1. lines 85-87: Endothelial cells rely more on glycolysis than OXPHOS. A higher dependence of OXPHOS leads to increase in free radicals which perpetuates endothelial damage. This is the preferred pathway, inspite of OXPHOS resulting in a higher ATP turnover. It is important to describe the importance of glycolysis vs OXPHOS in endothelial cells, and how the imbalance plays a crucial role in endothelial senescence and damage. While the authors touch upon this, they are very brief with it and it is not at all fleshed out. Additionally, mitochondrial dysfunction an oxidative stress are well-known to play important roles in endothelial senescence. This has not been fleshed out much in the review at all. Please look at Haspula et al., 2019 where the importance of metabolic shift from glycolysis and OXPHOS, and its impact on angiogenesis, is presented in more detail.
2. There is not much discussed about mechanical stimuli or disturbed flow in your review. Please refer to a recently published review by Sun and Feinberg (2021)
3. While you mention about inflammation in the context of various diseases, you have not mentioned about various cytokines (IL6, Tnf-a) and even inflammasome (Nlrp3) that have been implicated in endothelial senescence. This deserves a separate section.
4. You mention 'Angiotensin' as a separate section. But this should be RAAS and not just Angiotensin. There is Angiotensin 2, Angiotensin 3, Angiotensin 4 and Ang 1-7. This section is also not explained in great detail, and needs more information about downstream effectors of AT1R and the interplay between Mas and AT1R.
Round 2
Reviewer 1 Report
- Please look abstract carefully. There are some errors like duplications in lines 15-18.
- I believe that the aim of this review manuscript is to discuss mechanisms of EC senescence to find therapeutic targets for age-associated diseases. Thus, it would be better to provide any evidence that shows the beneficial or promising effects of each mechanism discussed in the manuscript on aging animal or human models, if it is possible.
- The authors defined that senescent ECs display flattened morphology. What’s the flattened morphology?
- In line 91, what’s the meaning of “proliferating HUVEC”? Is this normal or young HUVEC? or treated HUVEC?
- In line 101, the authors describe oxidative stress and ROS. If authors can discuss EC-specific ROS sources compared to other cell types, that would be helpful for readers to understand EC-specific phenotype.
- Generally, the review article is so much descriptive. Please put some direction of research in conclusions.
Reviewer 2 Report
The authors have responded adequately to all my comments. No further comments
Author Response
We appreciated Reviewer’s valuable comments in round 1.
Round 3
Reviewer 1 Report
Dear authors,
Thank you for addressing each comments and questions.